# Sex-Specific Differences in Lysine, 3-Hydroxybutyric Acid and Acetic Acid in Offspring Exposed to Maternal and Postnatal High Linoleic Acid Diet, Independent of Diet

**DOI:** 10.3390/ijms221910223

**Published:** 2021-09-23

**Authors:** Nirajan Shrestha, Steven D Melvin, Daniel R. McKeating, Olivia J. Holland, James S. M. Cuffe, Anthony V. Perkins, Andrew J. McAinch, Deanne H. Hryciw

**Affiliations:** 1School of Medical Science, Griffith University, Southport, QLD 4222, Australia; nirajan.shrestha@griffithuni.edu.au (N.S.); dm942@cam.ac.uk (D.R.M.); o.holland@griffith.edu.au (O.J.H.); a.perkins@griffith.edu.au (A.V.P.); 2Australian Rivers Institute, School of Environment and Science, Griffith University, Brisbane, QLD 4111, Australia; s.melvin@griffith.edu.au; 3School of Biomedical Science, The University of Queensland, Brisbane, QLD 4061, Australia; j.cuffe1@uq.edu.au; 4Institute for Health and Sport, Victoria University, Melbourne, VIC 8001, Australia; andrew.mcainch@vu.edu.au; 5Australian Institute for Musculoskeletal Science (AIMSS), Victoria University, Melbourne, VIC 8001, Australia; 6School of Environment and Science, Griffith University, Nathan, QLD 4111, Australia; 7Centre for Planetary Health and Food Security, Griffith University, Nathan, QLD 4111, Australia

**Keywords:** linoleic acid, sex, metabolite, offspring

## Abstract

Background: Linoleic acid (LA) is an essential polyunsaturated fatty acid (PUFA) that is required for foetal growth and development. Excess intake of LA can be detrimental for metabolic health due to its pro-inflammatory properties; however, the effect of a diet high in LA on offspring metabolites is unknown. In this study, we aimed to determine the role of maternal or postnatal high linoleic acid (HLA) diet on plasma metabolites in adult offspring. Methods: Female Wistar Kyoto (WKY) rats were fed with either low LA (LLA) or HLA diet for 10 weeks prior to conception and during gestation/lactation. Offspring were weaned at postnatal day 25 (PN25), treated with either LLA or HLA diets and sacrificed at PN180. Metabolite analysis was performed in plasma samples using Nuclear Magnetic Resonance. Results: Maternal and postnatal HLA diet did not alter plasma metabolites in male and female adult offspring. There was no specific clustering among different treatment groups as demonstrated by principal component analysis. Interestingly, there was clustering among male and female offspring independent of maternal and postnatal dietary intervention. Lysine was higher in female offspring, while 3-hydroxybutyric acid and acetic acid were significantly higher in male offspring. Conclusion: In summary, maternal or postnatal HLA diet did not alter the plasma metabolites in the adult rat offspring; however, differences in metabolites between male and female offspring occurred independently of dietary intervention.

## 1. Introduction

Maternal nutrition plays a critical role in foetal development and offspring outcomes. Previous studies have shown that alteration in dietary quality has a significant impact in the foetal environment, which corresponds with alteration in metabolic parameters later in life [1,2]. High fat diet (HFD) during pregnancy has been reported to affect the metabolic health of offspring [3]; however, the role of specific fatty acids during pregnancy on offspring metabolic health is poorly understood. Linoleic acid (LA) is an essential polyunsaturated fatty acid (PUFA) which is important for foetal growth and development. Even though LA is essential in our diet, excessive consumption of LA may be detrimental for health [4,5]. In Western societies, increased dietary availability has led to greater consumption of LA over the years [6], with intake of LA increasing to three times the recommended daily intake in the last few decades [7]. Elevated LA is associated with increased production of arachidonic acid (AA) which can be metabolised into lipid mediators, including pro-inflammatory eicosanoids, and prostaglandins [8]. Further, LA can increase endogenous endocannabinoids in rodent models [9,10]. Additional animal studies have demonstrated that elevated LA increased the production of pro-inflammatory cytokines [11]. Randomized control trials have also shown the detrimental effect of elevated LA on human health [12,13]. However, there are limited data studying the effect of high LA diet during pregnancy on offspring health. Recently, we have demonstrated that high maternal LA affects offspring cardiac function [14] and metabolic parameters [15] in a rat model.

Metabolomics, the study of small molecules and their interactions within a biological system, has unique potential for providing information about individual biological status arising from genetic, and environmental interaction, whilst also determining prospective biomarkers that may predict the incidence and severity, as well as the progression of disease. [16]. Plasma metabolomics provides information about individual biological status arising from genetic and environmental interaction [17]. Previous studies have shown the correlation between various metabolites and incidences of metabolic disorders such as impaired glucose tolerance and insulin resistance [18,19]. While most of the previous studies performed in the field of metabolomics were conducted in adults, recent studies measuring metabolites in pregnant women were useful for predicting metabolic outcomes in offspring [16]. Studies have also shown that maternal conditions, such as high body mass index (BMI) and under nutrition, during pregnancy affect the foetal metabolome [20,21]. A meta-analysis showed that maternal BMI is associated with cord blood metabolites (specifically, the branched chain amino acids; BCAAs), and that these correlated with a newborn metabolic signature characteristic of metabolic disorders in adults [21]. 

The use of plasma metabolomics may provide insight into established and novel determinants that promote the development of disease. For example, nutrition and gut microbiome composition may be linked to a metabolomic profile that is indicative of disease pathology or highlight the benefits of any treatment strategies. Metabolomics has been used to identify adverse metabolite profiles associated with elevated concentrations of LA [22], including profiles indicative of metabolic stress [23]. Metabolomic studies in pregnant women have been used for the evaluation of foetal development and pregnancy-related complications [24,25]. However, the effect of maternal LA diet during pregnancy on offspring metabolites has not been studied. Numerous animal studies have shown sex specific foetal programming in the response to maternal nutrition [26,27]. Metabolic diseases such as diabetes mellitus and cardiovascular disease are influenced by sex [28]. Metabolic profiling in the rat tissues has shown the influence of sex on metabolites that provide the important information for the explanation of the basis of the sex difference observed in physiological conditions [29]. 

This study aims to assess metabolites using Nuclear Magnetic Resonance (NMR) to investigate the effect of maternal or postnatal diet high in linoleic acid on plasma metabolites in adult rat offspring. 

## 2. Results

### 2.1. Effect of Maternal and Postnatal High Linoleic Acid Diet on Plasma Metabolites in Adult Offspring

The maternal and postnatal HLA diets do not have any overarching effect on plasma metabolites in male (Figure 1) offspring as observed in heatmaps. Principal component analysis (PCA) determines how diet affects the clustering of different treatment groups, which illustrated there was no specific clustering among the groups in male (Figure 2) offspring. Maternal and postnatal HLA diets do not have any overarching effect on plasma metabolites in female offspring (Figure 3) and PCA analysis identified no specific clustering among the groups in female (Figure 4) offspring. 

### 2.2. Clustering among Male and Female Offspring

Sparse PLS Discriminant Analysis (sPLS-DA) of all the individual samples demonstrated that there was clustering among the male and female groups, independent of maternal or postnatal dietary intervention (Figure 5). The results indicated that the samples in the male and female groups had distinctively different metabolomic markers when using the top ten metabolites. These differences where further highlighted in the volcano plot, which showed that lysine (chemical shift 1.72 ppm and 3.02 ppm) was significantly higher in female rats, by at least a factor of 1.5 (Figure 6A), whilst 3-hydroxybutyric acid, and acetic acid were significantly higher in males, also by a factor of 1.5 (Figure 6A). The difference in lysine concentrations was further highlighted in Figure 6B,C, where they were compared via ROC curve, indicating that lysine provided a 96.5% (CI 89.6–99.2%) ability to discriminate between male and female offspring, regardless of maternal or postnatal diet.

## 3. Discussion

Metabolomics measures large sets of endogenous small molecules/metabolites in a biological system and provides a powerful phenotyping tool [30]. It has been used for discovery of biomarkers for metabolic dysfunction and provides better understanding of the metabolic pathway of diseases [31]. Very few metabolomic studies have been conducted in the field of pregnancy and foetal programming [32,33] Maternal nutrition during pregnancy and lactation modifies the metabolic phenotype of the offspring [34]. This study explored the role of maternal and postnatal diet high in LA on offspring plasma metabolites. 

In our previous study, we reported that maternal and postnatal diet high in LA alters metabolic parameters, mainly circulating leptin and cholesterol, in adult offspring in a sex-specific manner [15]. In this study, there is no alteration in measured metabolites among the dietary groups, so that maternal and postnatal LA had no effect on the distribution of metabolites in the adult offspring. Notably, we did identify that some metabolites differed considerably between sexes, but this was independent of the maternal or postnatal diet. Previous studies have demonstrated that the sex influences plasma metabolites [35,36]. In a longitudinal plasma metabolomic study, 68 metabolite trajectories significantly differed by sex, including sphingolipids, which tend to increase in women and decrease in men with age [35]. 

In the present study, there was clustering among male and female groups, as shown by sPLS-DA of all the individual samples, independent of dietary intervention. Lysine was significantly higher in female in comparison to male offspring. In yeast, lysine biosynthesis has the protective role against linoleic acid hydroperoxide-induced oxidative stress [37]. Increasing dietary intake of LA specifically increased LA peroxidation products [38]. This suggests that lysine production may have a protective role against oxidative stress induced by elevated LA. Despite predicted differences in oxidative stress between sexes with the expected LA peroxidation products due to excess LA in the diet, we did not see differences in lysine in offspring fed with HLA diet. This gender difference may suggest that females are more protected from any adverse programming effects of an elevated LA diet. Lysine, an essential amino acid, is thought to promote human development, enhance immune function, and improve the function of central nervous tissue. Lysine is also an essential factor for bone collagen synthesis [39].

Based on the heatmaps, considerable metabolic variability between individuals and differences between sexes were more pronounced and potentially overshadowed differences due to HLA diet. This is unexpected, as the genetic background and environment between the rats within the same groups are not altered. This variability may have resulted in differences due to diet being masked by the population diversity. Notably, we have in previous studies identified differences in circulation fatty acids and hepatic genes [13], which suggests that this diversity is unlikely to mask underlying metabolite diversity.

The most common analytical platforms used for untargeted metabolomics studies are liquid- or gas-chromatography paired with mass spectrometry (LC-MS and GC-MS, respectively), or Nuclear Magnetic Resonance (NMR) spectroscopy. There are advantages and disadvantages to each platform, which have been discussed and debated elsewhere (Emwas, 2019 #4435) and can be largely condensed down to greater coverage of metabolites by MS countered by better high-throughput capability, greater reproducibility and reduced costs for NMR [40]. The present study used NMR as this was a preliminary exploratory study looking for potential effects of LA on the rat metabolome, identifying a small number of metabolite differences between sexes. Future investigation into the observed differences may benefit from the greater sensitivity of LC- or GC-MS, which may identify further metabolites of interest and allow for functional pathway analysis to better understand how males and females differ.

In this study, 3-hydroxybutyric acid and acetic acid were significantly higher in male in comparison to female offspring. This difference is independent of diet. 3-hydroxybutyric acid, acetone, and acetoacetic acid, are ketone bodies that are a group of organic compounds of intermediary fat metabolism. High concentrations of ketone bodies decrease the rate of β-oxidation of fatty acids [41]. In humans, circulating 3-hydroxybutyric acid concentrations greater than 3 mmol/L in children or 3.8 mmol/L in adults are considered the clinically significant level in diabetic ketoacidosis [42]. It has previously been shown that women have higher postprandial ketone bodies than men, which may reflect increased postprandial free fatty acid (FFA) oxidation in the liver [43]. This suggests that future studies should investigate the association between maternal LA intake, ketone bodies and hepatic fatty acid oxidation in offspring.

## 4. Materials and Methods

### 4.1. Experimental Animal Model and Diet

Eight-week-old female Wistar Kyoto (WKY) rats were purchased from the Australian Resource Centre (Perth, WA, Australia) and housed in accordance with the Australian Code of Practice for Care and Use of Animals for Scientific Purpose. Animal ethics was approved by the Griffith University Animal Ethics Committee (NSC/01/17/AEC). Female rats were housed in individually ventilated cages under 12 h light–dark cycle at a temperature of 20–22 °C and provided with standard food pellets during a week of acclimatisation and tap water ad libitum throughout the study. After acclimatization, rats were randomised to consume either a low LA (LLA: 1.44% of energy from LA, *n* = 8) or high LA (HLA: 6.21% of energy from LA, *n* = 8) diet for 10 weeks. The composition of the animal diet has been previously detailed [44]. These diets were isocaloric and matched for total fat and total n-3 PUFA contents, and the concentration of LA in the diet reflects the average Australian diet [7]. After 8 weeks of dietary treatment, vaginal impedance was measured daily for at least two oestrous cycles using a rat vaginal impedance checker (Muromachi Kikai Co. Ltd., Tokyo, Japan). Rats were considered ready for mating after 10 weeks of dietary treatment and when vaginal impedance was 4.5 × 10^3^ Ω or greater and at this time female rat was placed with a Wistar Kyoto male rat overnight. Respective diets were provided ad libitum during gestation as well as the lactation period. Offspring from mothers were weaned at post-natal day 25 (PN25). From weaning, offspring were housed with one or two other litter mates of the same sex and fed either LLA or HLA diets. The rat offspring were sacrificed at PN180 (6-month-old adult offspring). Offspring were terminally anesthetised with an intraperitoneal injection of sodium pentobarbital (60 mg/kg). Blood samples were collected by cardiac puncture, centrifuged at 5000× *g* for 10 min to separate plasma and stored at −80 °C for analysis.

### 4.2. Sample Analysis

Blood plasma was processed using a modified Folch/Bligh–Dyer extraction protocol to separate and isolate small polar molecules and lipids from protein and cellular debris. The general protocol for metabolite extraction has been described in several recent publications [45,46]. Briefly, 800 μL ice-cold methanol was added to a 400 μL aliquot of blood plasma and the sample was ultra-sonication using a Vibra-Cell VCX-130 probe sonicator (Sonics and Materials Inc., Newtown, CT, USA). The samples were incubated at −20 °C for 24 h, after which 1600 μL chloroform and 300 μL ultrapure water were added and the samples were vortexed and centrifuged for 10 min (16,000× *g* at 4 °C). The upper hydrophilic (polar) and lower hydrophobic (lipid) phases were carefully partitioned into separate glass amber vials and stored at −80 °C until analysis.

Extracted fractions were dried using a Genevac HT-12 Series 3i centrifugal evaporator (Genevac Technologies, Suffolk, England). Polar extracts were reconstituted in 200 µL phosphate buffer made with deuterium oxide (D_2_O), with 0.05% sodium-3-(trimethylsilyl)-2,2,3,3-tetradeuteriopropionate (TSP) as an internal reference. Lipid extracts were reconstituted in 200 µL chloroform-*d* (CDCl_3_) with 0.05% tetramethylsilane (TMS) as an internal reference. Individual reconstituted samples were transferred to 3 mm NMR tubes using a Hamilton^®^ zero dead volume glass syringe (Hamilton Company, Nevada, USA) and analysed on an 800 MHz Bruker^®^ Avance III HDX Nuclear Magnetic Resonance (NMR) spectrometer. The NMR is equipped with a Triple (TCI) Resonance 5 mm Cryoprobe with Z-gradient and automatic tuning and matching. Spectra were acquired at 298° K with D_2_O or CDCl_3_ used for field locking and TSP or TMS (^1^H δ 0.00, ^13^C δ 0.00) as internal references. Proton (^1^H) spectra were acquired for all samples using zg30 pulse program with 32 scans, 0.8 s relaxation delay, 8.20 μs pulse width and a spectral width of 16 kHz (^1^H δ –3.75–16.28). Edited ^1^H-^13^C Heteronuclear Single Quantum Coherence (HSQC) spectra were acquired for representative samples with 128 scans and 128 experiments, 0.8 s relaxation delay, 8.20 μs pulse width and spectral widths of 12.8 kHz (^1^H δ –3.23–12.79) and 33.1 kHz (^13^C δ –9.40–155.2). Zg30 was utilised in place of noesygppr1d pulse programs as the samples were reconstituted in D_2_O, and the residual water peak was minor and did not overlap with any critical spectral features (metabolites) [45]. 

^1^H spectra were manually phase corrected, automatically baseline adjusted (ablative) and referenced and normalised to the internal reference (TSP or TMS at ^1^H, δ 0.00) using MestReNova version 11.0.4 (Mestrelab Research S.L.). After processing, the spectra were stacked (superimposed), the integrated area of each identified feature was calculated using the advanced data analysis tool in MestReNova, and this data was exported to Excel for statistical analysis.

Polar metabolites were identified by first fitting the ^1^H spectra using Chenomx v8.5 software (Chenomx Inc., Edmonton, AB, Canada), then confirming the identifications by comparing HSQC spectra with reference spectra in the Human Metabolome Database. Lipids (i.e., protons corresponding to head groups of different lipid classes) were identified with reference to existing publications describing NMR-based lipidomics and by applying standard methods for determining the structure of organic compounds [47,48,49].

### 4.3. Statistical Analysis

R ×64 4.0.2, and MetaboAnalyst 5.0 were used to perform further exploratory and biomarker analysis [50]. Heatmaps allowed for graphic depiction of the large datasets generated by metabolomic analysis, simplifying understanding. When heatmaps were coupled with Pearson’s dendrograms for metabolite, it enabled the ability to visualise metabolite clusters illustrating their distance, or dissimilarity between clusters. Principal component analysis (PCA) and Sparse Partial least square discriminate analysis (PLS-DA) was performed to visualise non-discriminate and discriminant groupings within the data, respectively. Volcano plots were generated using t-tests as well as fold changes with significance set at *p* < 0.05, and 1.5-fold change, respectively. Receiver operating characteristic (ROC) curve used to determine the ability of metabolites to discriminate between two groups. 

## 5. Conclusions

In conclusion, maternal and postnatal diet high in LA did not alter plasma metabolites in the adult rat offspring. However, metabolites, specifically lysine, 3-hydroxybutyric acid and acetic acid were altered among male and female offspring independent of diet. Future studies should emphasize the utilization of metabolomics to investigate the metabolites involve in the fatty acid oxidation. 

## Figures and Tables

**Figure 1 ijms-22-10223-f001:**
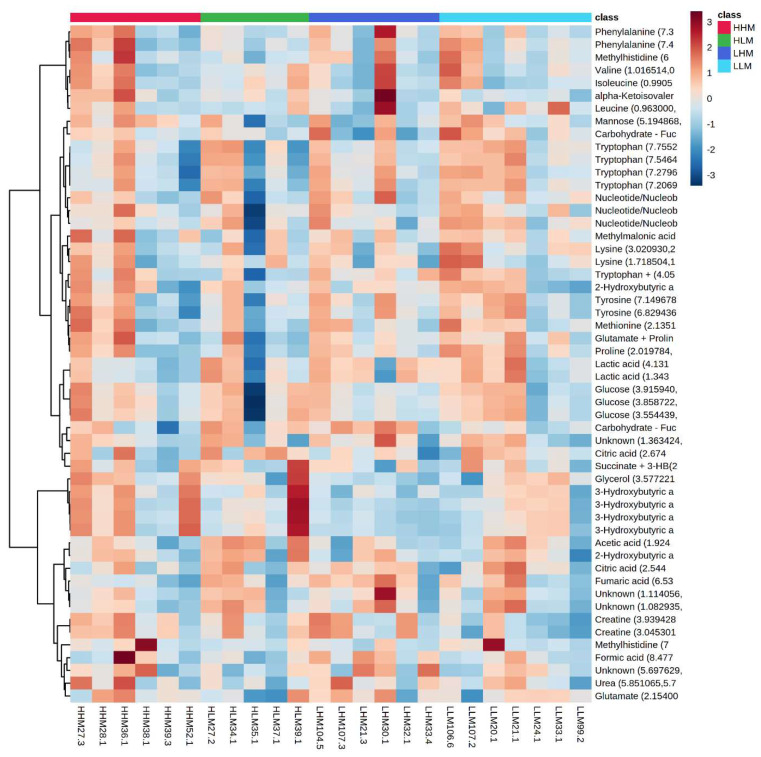
Heatmaps and dendrograms of metabolites in offspring exposed to high maternal and high postnatal LA diets in males (HHM: red), high maternal and low postnatal LA diets in males (HLM: green), low maternal and high postnatal LA diet in males (LHM: dark blue), and low maternal and low postnatal LA diet in males (LLM: light blue). All metabolites have been logarithmically scaled. Heatmap scales are denoted by red being higher concentrations, blue being lower, with white residing in the middle. Pearson clustering algorithm used for dendrograms to indicate relationships between metabolites in the cohort.

**Figure 2 ijms-22-10223-f002:**
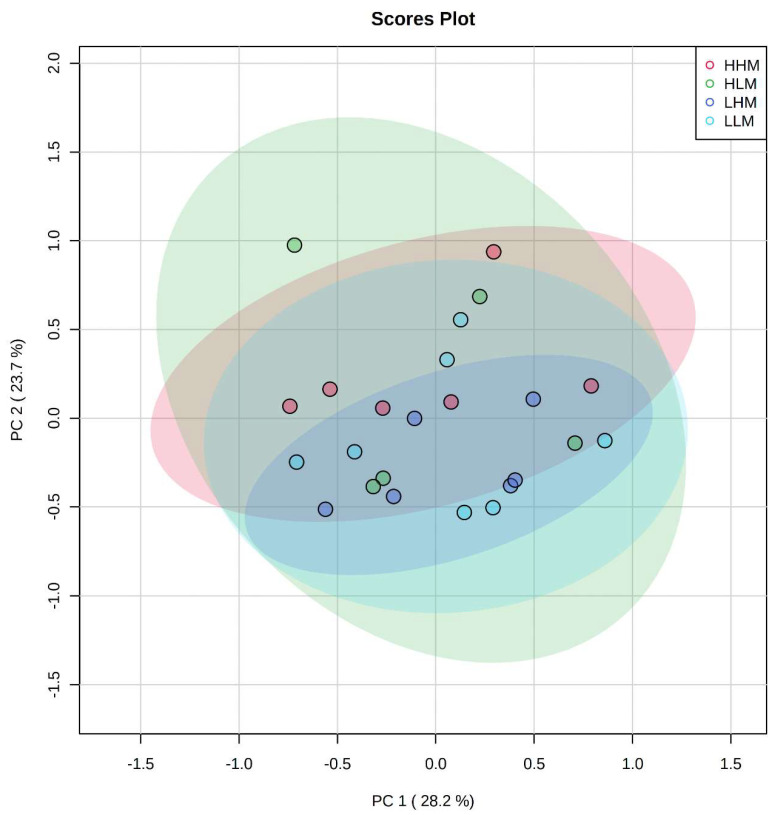
PCA of male offspring, indicating a lack of clustering between offspring exposed to high maternal and high postnatal LA diets in males (HHM: Red), high maternal and low postnatal LA diets in males (HLM: green), low maternal and high postnatal LA diets in males (LHM: dark blue), and low maternal and low postnatal LA diets in males (LLM: light blue). All metabolites have been scaled logarithmically. Illustrates variance (%) in PC1 and PC2.

**Figure 3 ijms-22-10223-f003:**
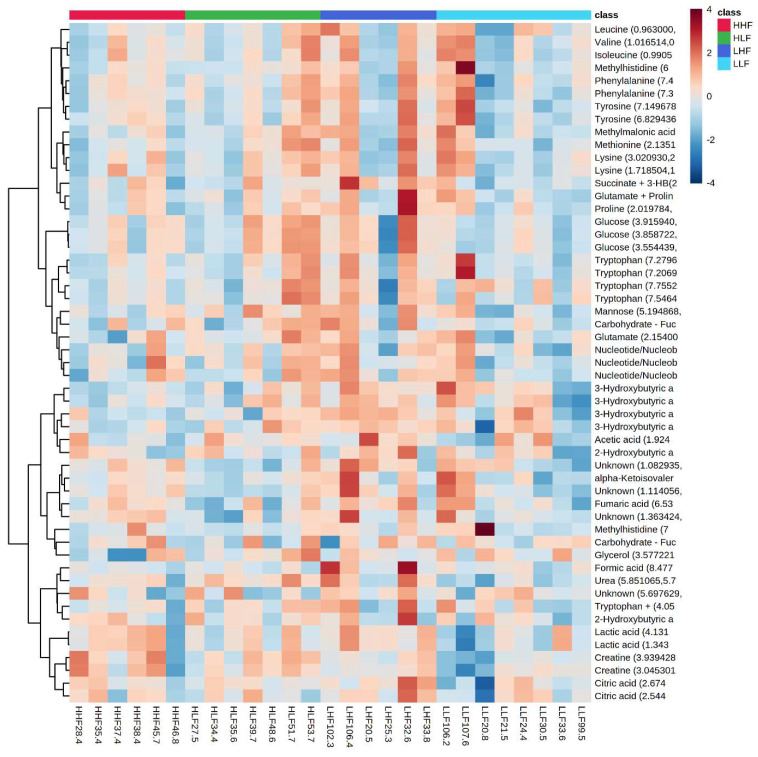
Heatmaps and dendrograms of metabolites in offspring exposed to high maternal and high postnatal LA diets in females (HHF: red), high maternal and low postnatal LA diets in females (HLF: green), low maternal and high postnatal LA diets in females (LHF: dark blue), and low maternal and low postnatal LA diets in females (LLF: light blue). Offspring numbers are indicated on the bottom axis of the graph, whilst the top axis indicates their corresponding group colors. All metabolites have been logarithmically scaled. Heatmap scales are denoted by red being higher concentrations, blue being lower, with white residing in the middle. Pearson clustering algorithm used for dendrograms to indicate relationships between metabolites in the cohort.

**Figure 4 ijms-22-10223-f004:**
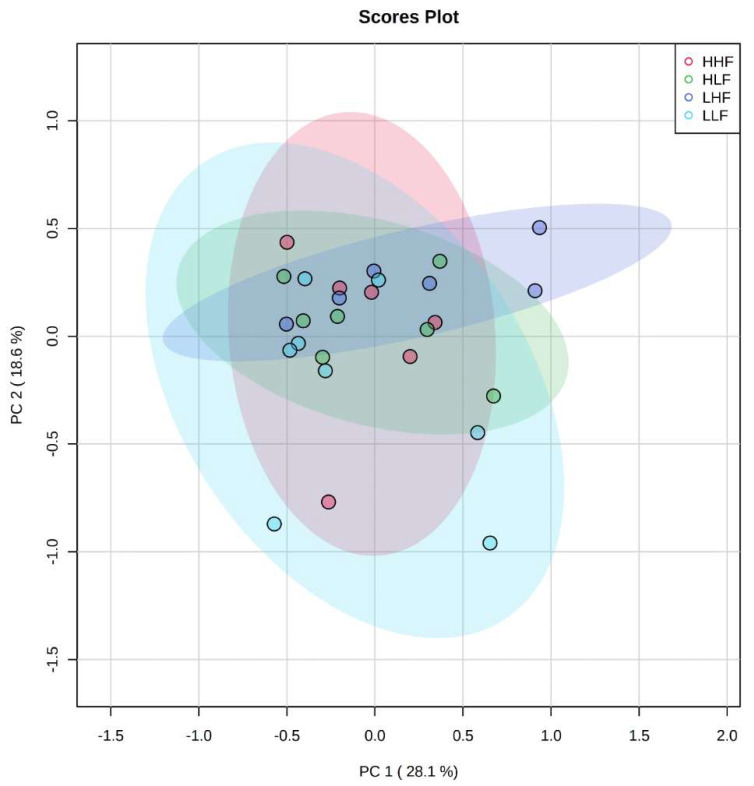
Heatmaps and dendrograms of metabolites in offspring exposed to high maternal and high postnatal LA diets in females (HHF: red), high maternal and low postnatal LA diets in females (HLF: green), low maternal and high postnatal LA diets in females (LHF: dark blue), and low maternal and low postnatal LA diets in females (LLF: light blue). Offspring numbers are indicated on the bottom axis of the graph, whilst the top axis indicates their corresponding group colors. All metabolites have been logarithmically scaled. Heatmap scales are denoted by red being higher concentrations, blue being lower, with white residing in the middle. Pearson clustering algorithm used for dendrograms to indicate relationships between metabolites in the cohort.

**Figure 5 ijms-22-10223-f005:**
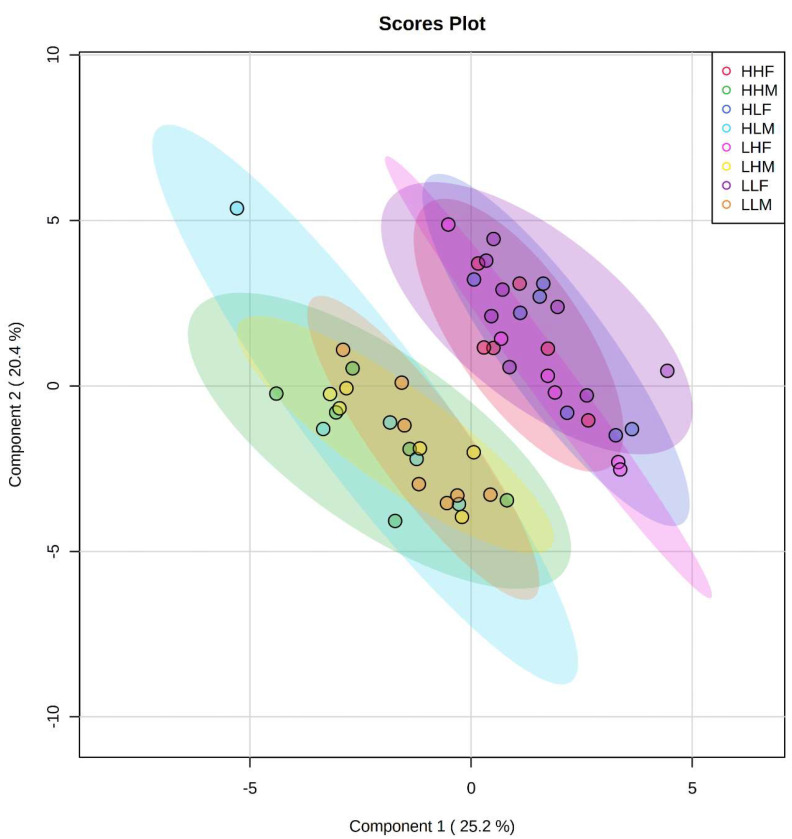
sPLS-DA analysis of all groups. HHF (red), HHM (green), HLF (blue), HLM (light blue), LHF (pink), LHM (yellow), LLF (purple), LLM (orange). Number of variables per component limited to 10 across all, only first 2 components (% variance) illustrated. All metabolites are logarithmically transformed. Indicated the discriminate clustering of male and female groups.

**Figure 6 ijms-22-10223-f006:**
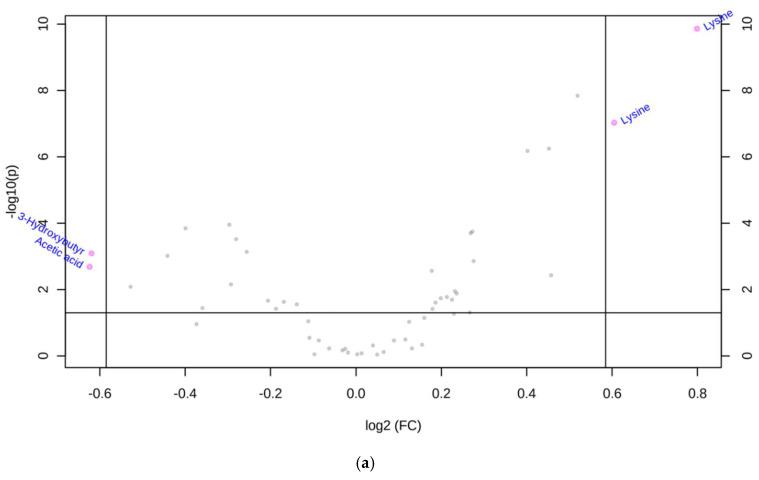
Volcano-plot, ROC curve, and column graph of male and female samples, groups regardless of maternal or foetal dietary influence. (**a**) Volcano plot comparison of fold changes against t-test p-values of significance in direction of F/M where negative fold changes (log2 (FC)) indicates lower in females, positive fold changes indicate lower in males. Threshold for fold changes set at 1.5 times (3-Hydroxybutyr: 3-Hydroxybutyric Acid). (**b**) Receiver operating characteristic curve for lysine, indicates the false positive rate for the metabolite against the true-positive rate for discriminating against male and female offspring. (**c**) Column graph of lysine concentrations in male and female samples, red line indicates the threshold for splitting samples in the ROC curve (**b**).

## Data Availability

The data presented in this study are available on request from the corresponding author.

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
