# Peer review of "Sex-Specific Differences in Lysine, 3-Hydroxybutyric Acid and Acetic Acid in Offspring Exposed to Maternal and Postnatal High Linoleic Acid Diet, Independent of Diet"

_ijms, 2021, doi:10.3390/ijms221910223_

Round 1
Reviewer 1 Report
The authors have investigated if low and high LA could be detrimental to fetal health. NMR has been used to understand the global changes in the metabolome. The scientific question is very interesting. However, please correct the following issues.
Please address the following issues
Elevated LA is associated with increased production of arachidonic acid (AA) which can be metabolised into lipid mediators, including pro-inflammatory eicosanoids, and prostaglandins [8].
1.Elevated LA is also known to increase in endocannabinoids in host. Please mention this and cite following papers.
Linoleic acid in diets of mice increases total endocannabinoid levels in bowel and liver: modification by dietary glucose, Ghosh et.al., 2019, Obesity, Science & Practice.
Dietary linoleic acid elevates the endocannabinoids 2-AG and anandamide and promotes weight gain in mice fed a low fat diet, Alvheim AR et.al., 2014, Lipids.
2.The figures legends are not complete. Please write the full forms of “HHM”, “HLM”, “LHM”, “LLM”, “HHF”,”HLF”,”LHF”,”LLF”.This should be also mentioned in text.
3.The authors have used 6%LA in the diets. What was the rationale for using that? 8% LA is what is generally used for high LA administration. Please mention the composition of the diets in a table and a rationale for using 6%LA.
4. The authors used zg30 pluse program, what was the rationale for this? Metabolomics experiment generally uses noesygppr1d experiment. Mention the rationale and cite a paper that uses it.
5.How did the authors normalize the data?
6.Figure 1 and 2 can be combined as one figure, so is Figure 3 and 4.
7.The authors have concluded based on NMR data, however NMR cannot pick up low abundant molecules in the system and perhaps LC/MS/MS profiling is useful in this case. The drawbacks of the study should be mentioned in the text and as a scope for future experiment LC/MS/MS could be mentioned.
Reviewer 2 Report
The authors studied the role of maternal or postnatal high linoleic acid diet on plasma metabolites in male and female adult offspring and indicated that the diet did not alter plasma metabolites. It was revealed clustering among male and female offspring independent of maternal and postnatal dietary intervention, such as Lysine was higher in female offspring, while 3-hydroxybutyric acid, and acetic acid were higher in male offspring.
Manuscript is clearly written and data is clearly represented, but in my opinion there are some fundamental mistakes in showing the study.
In title is reported: “Sex-specific differences in metabolites in offspring exposed to maternal and postnatal high linoleic acid diet is independent of diet” and I think it is necessary to specify in the title which metabolites are concerned and also in introduction and in discussion it is necessary to better explain why this category of metabolites were analysed and their involvement with high linoleic acid diet.
Moreover, the sex-specific differences are independent of the diet so it is necessary to better discuss these characteristics between the sexes and what possible consequences they could have for example greater protection for ...?
The authors can modify the manuscript and subsequently I can reconsider it.
Round 2
Reviewer 1 Report
The authors have responded to all the points raised.
Reviewer 2 Report
I thank the authors for comprehensive feedback and for me there is only a modification to add in title: in line 4 is more correct to write “are independent of the diet”.